# Esophageal Cancer with Early Onset in a Patient with Cri du Chat Syndrome

**DOI:** 10.3390/diseases12010009

**Published:** 2023-12-29

**Authors:** Cesare Danesino, Monica Gualtierotti, Matteo Origi, Angelina Cistaro, Michela Malacarne, Matteo Massidda, Katia Bencardino, Domenico Coviello, Giovanni Albani, Irene Giovanna Schiera, Alexandra Liava, Andrea Guala

**Affiliations:** 1Department of Molecular Medicine, University of Pavia, 27100 Pavia, Italy; 2Scientific Committee of A.B.C. Associazione Bambini Cri du Chat, 50026 Firenze, Italy; angelinacistaro06@gmail.com (A.C.); gualaandrea0@gmail.com (A.G.); 3UOC Chirurgia Generale Oncologica e Mininvasiva, Ospedale Niguarda, 20162 Milano, Italy; monica.gualtierotti@ospedaleniguarda.it (M.G.); matteo.origi@ospedaleniguarda.it (M.O.); 4Nuclear Medicine Unit, Salus Alliance Medical, 16129 Genova, Italy; 5Pediatric Study Group Italian Associaton of Nuclear Medicine (AIMN), 20159 Milan, Italy; 6UOC Laboratorio di Genetica Umana, IRCCS G. Gaslini, 16147 Genova, Italy; michelamalacarne@gaslini.org (M.M.); matteo.massidda@gmail.com (M.M.); domenicocoviello@gaslini.org (D.C.); 7Department of Medical, Surgery and Experimental Sciences, University of Sassari, 07100 Sassari, Italy; 8Niguarda Cancer Center, Grande Ospedale Metropolitano Niguarda, 20162 Milano, Italy; katia.bencardino@ospedaleniguarda.it; 9Department of Neurology and Neurorehabilitation for Severe Brain Injures Acquired Ospedale Moriggia Pelascini, 22015 Gravedona, Italy; g.albani@auxologico.it; 10Independent Researcher, 90121 Palermo, Italy; schierairene85@gmail.com; 11UOC Neuropsichiatria Infantile, 22100 Como, Italy; alexandra.liava@ti.ch; 12UOC Pediatria, Ospedale Castelli, 28921 Verbania, Italy

**Keywords:** Cri du Chat syndrome, esophageal adenocarcinoma, whole exome analysis, brain PET

## Abstract

Background: In Cri du Chat (CdC), cancer as comorbidity is extremely rare. In databases from Denmark, Spain, Australia, New Zealand, and Japan, no cancer was reported; in Italy and Germany, four cancers were identified out of 321 CdCs. Methods: In a 29-year-old CdC patient, clinical investigations following hematemesis led to the diagnosis of esophageal adenocarcinoma (EAC). A high pain threshold was also observed. Conventional and molecular cytogenetic defined the size of the deletion, and exome analysis on the trio completed the molecular work. Results: Cytogenetic analysis showed a de novo chromosomal alteration: 46,XY,ishdel(5)(p14.3)(D5S28-) and arr[GRCh37] 5p15.33p14.3(1498180_19955760)x1. A quantitative sensory test demonstrated a high heat threshold. A 18f-fluorodeoxyglucose PET/TC scan of the brain failed to detect reduction of metabolism in the somatosensory area or insular cortex. Exome analysis in the trio (patient and parents) failed to identify variants to be interpreted as a likely risk factor for EAC. Conclusion: We conclude that the presence of well-known risk factors (maleness, obesity, gastroesophageal reflux, and Barrett’s metaplasia) in a patient with very limited capability of expressing discomfort or referring clinical symptoms have been the main risk factors for developing EAC. At present, based on the available data, there is no evidence of any increased risk of developing cancer in CdC patients.

## 1. Introduction

Cri du Chat (CdC) is a rare syndrome caused by deletions of various size in the short arm of chromosome 5, and developmental delay is the main problem observed [1]. No major malformations are usually associated with cognitive problems, and among comorbidities, cancer, of any type, has been reported occasionally in a few cases. In the Danish population, 32 comorbidities were identified in 62 CdCs (aged 6.28 ± 18.24), but esophageal cancer or, in general, any type of cancer was not reported [2]; similarly, in a series of 70 Spanish cases (aged 8.99 ± 8.94, range 0.1–45) [3], in a group of 102 unpublished patients from Australia and New Zealand (age 10 months–40 years) [4], and in a group of 111 Japanese patients [5], no cancer was reported, as far as we are aware. In a recent paper on Italian and German databases [6], 4 patients with different types of cancer were identified out of 321 CdCs. It should be noted that in this group of cases, 18.7% of patients were over 40 years, with the oldest patient being 74 years old. The same authors noted that for three of the cancers reported (thyroid papillary carcinoma, gastric carcinoid, and breast cancer), the age of onset was clearly earlier as compared to the expected age of occurrence of the same type of cancer in the general population. They concluded that although, at present, there is no evidence of an increased risk of developing cancer in CdCs, reporting even sporadic cases will expand our knowledge of the relationship, if any, between CdC and cancer. No new cases have since been added to the German database [7], while one new CdC patient affected with cancer, the one in this report, was identified in Italy.

With this aim, we report a new case of esophageal cancer in a 29-year-old CdC patient.

## 2. Case Report

The patient, a male, was born in 1991; the pregnancy was uneventful up the 34th week, when a cesarean section was performed as an emergency due to acute foetal distress. The APGAR score was 3–4 at 1 and 5 min, the birth weight was 2610 g, the length was 44 cm, and the head circumference was 32.5 cm. The parents (father 30 yrs old, mother 24 yrs old at time of birth), unrelated and healthy, originate from Lombardia, North Italy; a brother (born in 1989) is alive and healthy. The patient was required to be immediately admitted to the intensive care unit because of neonatal asphyxia and was on mechanical ventilation for 5 days. Due to poor sucking, the newborn was fed with adapted milk through a nasogastric feeding tube. A heart ultrasound revealed a small ventricular septal defect, which underwent spontaneous resolution. The presence of some dysmorphic features (epicanthal folds, bitemporal narrowing, and a bilateral simian crease), severe hypotonia, and a typical cat cry suggested the presence of Cri du Chat syndrome. The diagnosis was then confirmed through routine chromosomal analysis and in situ hybridization (ish): 46,XY,ishdel(5)(p14.3)(D5S28-). The chromosomal alteration is de novo as the parental karyotype, including the FISH analysis, was normal. A brain MRI at the age of 2 months did not show any major abnormality. Gastroesophageal reflux (GER) was diagnosed at 1 month because of a typical clinical picture, and endoscopy demonstrated normal esophageal walls until the distal third, where hyperemia was present with a Z-line 1 cm above the gastroesophageal junction; the cardias was gaping.

Severe developmental delay was immediately clear: he pronounced his first words at the age of 18 months, he started walking at 3 years of age, he started running at 5 years of age, and toilet training was fully achieved at 9 years. He was able to build a tower with three cubes at the age of 4 yrs and with four cubes at age 7; he was able to dress autonomously at the age of 12 yrs. In the following years, he underwent extensive rehabilitation programs, with good results.

Physical development was similarly delayed; at the last control, at the age of 27 yrs, his height was 163 cm (50 °C) and his head circumference was 54.5 (>98 °C), with a centile specific for CdC patients [8]. The patient was obese, with a weight of 89 kg (+6SD), and the weight increase started at age 12.

Additional anthropometric data were compared to the general population [9]: his ear length was 6.8 cm (90 °C) with bilateral lobe hypoplasia; the inner canthal distance was 3.2 cm (75 °C); the outer canthal distance was 9 cm (50–75 °C); the total hand length was 17.5 cm, and the middle finger length was 7.5 cm (both at 50 °C); a single palmar crease was present bilaterally; the internipple distance was 26 cm (+3SD); and his chest circumference was 112 cm (+4SD). His foot length was 25 cm (10 °C), with bilateral clinodactyly.

Facial dysmorphic features included bitemporal narrowing, a bulbous nose, epicanthal folds, and an everted lower lip. Numerous dental caries were present. Mild kyphosis was also observed. The external genitalia were as in normal males. No additional major clinical problem was reported by the parents.

A neurological examination disclosed strabismus, with the left eye in mild abduction in the primary position; global dyspraxia, with difficulty in the coordination and execution of sequential movements while dressing; and the presence of ideomotor and constructional dyspraxia. The gait was mildly lurching, unstable, and carried out on a wide basis; the foot placement was irregular; he was unable to perform tandem gait; and there was also a difficulty in heel- and toe-walking.

We recorded dysmetria in the finger-to-nose test, difficulty in finger sequencing, dysdiadochokinesis, a strength of four in the upper and lower limbs [10], weakness slightly more pronounced on the right side, and slightly increased basal muscular tone. Fine coordination was partially conserved (he is able to play a pinball machine). In terms of reflexes, the upper limbs normally elicited, the patellar increased, the Achilles was difficult to elicit, the Babinski sign was negative, and there were no significant displacements at Mingazzini 1.

Sensory examination of the spinothalamic tract in the dermatome patterns explored is difficult, as the responses are not always accurate and consistent, but no unilateral deficits were identified. Oro-motor mild hypotonia and mild dyspraxia of the oro-lingual region were present, with a hoarse, nasal, and tremulous speech. Spontaneous speech was limited to sounds and a few contracted words. The patient used gestural language to facilitate communication; eye contact was good, and he usually managed to develop a good relationship with the examiner.

No apparent short-term recall deficits were observed, and his episodic memory was fairly good. Medium–severe cognitive retardation was evident with trial-and-error strategies, as well as difficulty in cause–effect associations and attention deficit.

Regarding EEG during wakefulness, there was a presence of rare spike-waves in the frontal–central–temporal regions, slightly predominant to the right, which were not related to any overt clinical sign and thus not requiring therapy. Behavioral observation with the Vineland adaptive behavior scale showed: communication 1 year 6 months, daily living skills 3 years 5 months, socialization 2 years 11 months, and motor abilities 2 years 10 months. He was partially autonomous in personal hygiene. He showed a tendency toward poor adherence to the rules and poor tolerance to frustration, with outbursts of anger and self-aggressiveness when upset. No alcohol or smoking habits were reported. The severely limited capability of the patient to report any discomfort or pai, was the cause of a reduced number of clinical investigations during his life. In particular, the patient was unable to (and did not) report any problem related to the esophagus or stomach when eating, so persistence or worsening of GER was not investigated and no endoscopic exploration was performed up to the moment of esophageal cancer diagnosis.

The parents stated that he apparently showed a high pain threshold, so a quantitative sensory test (QST) was performed, which demonstrated, as regards the cold stimulus, a normal threshold but a decidedly abnormally high tolerability (0 °C, normal range: 9.9–26.6 °C), while, concerning the heat, both the threshold (49.9 °C, normal range: 37.9–48.3 °C) and tolerability were higher than normal (50 °C, normal range: 39.1–46.7 °C) [11].

At the same time, aged 28 years old, the patient underwent a 18f-fluorodeoxyglucose (^18^F-FDG) Positron Emission Tomography (PET)/computed tomography (CT) scan of the brain to investigate alterations in brain glucose metabolism. This investigation was performed with the aim of detecting differences in brain ^18^F-FDG metabolism in CdC patients with different clinical presentations and to identify possible brain metabolic phenotypes of this syndrome [12].

Areas of significant hypometabolism were found in the left precuneal region/posterior cingulate cortex, in the fusiform gyrus (BA 31, 36), in the right body of the caudate nucleus, and in the posterior cerebellar lobe bilaterally. Other less severe areas of hypometabolism were found in the frontal–orbital cortex, bilaterally (BA 11, 47). No area of significant reduction of metabolism was detected in the somatosensory or insular cortex (Figure 1).

## 3. Cancer Clinical History

In December 2020, at age 29, an episode of hematemesis prompted further investigations, and after gastroscopy, a diagnosis of adenocarcinoma of the gastroesophageal junction developing over a Barret esophagus was performed. Stadiation with a CT scan and ultrasound endoscopy was in favor of CT3N+ M0 adenocarcinoma, for which the ideal treatment is perioperative chemotherapy associated with radical surgery.

Four cycles of Fluorouracil, Leucovorin, Oxaliplatin, and Docetaxel (FLOT) were given between January and March 2021, and a subtotal esophagectomy through laparoscopy and a thoracic approach (Hybrid Ivor Lewis procedure) was performed in April 2021. The post-operative course was uneventful, and the patient was able to eat normally and return home in ten days. An additional three cycles of FLOT were given in May–July 2021 (with peg-filgrastim); an episode of pneumonia was treated with piperacillin/tazobactam IV.

The histological diagnosis was tubular and papillar adenocarcinoma, intestinal type tumor regression grade (TRG) Ryan 3. Molecular analyses showed *HER2*- (IHC 1+); *MSS* (IHC pMMR); NGS FoundationOne CDx (Basket Study of Entrectinib (RXDX-101) for the Treatment of Patients With Solid Tumors Harboring *NTRK* 1/2/3 (Trk A/B/C), ROS1, or *ALK* Gene Rearrangements (Fusions) (STARTRK-2); Protocol GO40782: *CDKN2B* loss, *CDKN2A* loss, *MTAP* loss, *CDK6* amplification, *TP53:*R273C, *ACVR1B:*R456*, *APC:*E1464fs*8, *PIK3CA:*E542K. Pharmacogenomic profile: *DPYD* WT. Regular clinical follow-up occurred up to March 2022, when a relapse with metastatic localization in the liver and adrenal gland were observed. The patient died in May 2023.

## 4. Molecular Analysis

An array CGH was performed in 2020 (Figure 2), ISCN result: arr[GRCh37] 5p15.33p14.3(1498180_19955760)x1. The patient was part of a group of 10 CdC trios, on whom whole exome sequencing (WES), using routine methods, was performed within a research project on a genetic basis of clinical variation in CdC. Exome sequencing was performed on peripheral blood cells of the “trio” (patient and parents) in order to be able to verify, for any variant of interest, if it was inherited or de novo and the model of inheritance (recessive, compound heterozygous). The results were filtered for rare variants under models of autosomal recessive inheritance (homozygous or compound heterozygous) and de novo variants; the results are summarized in Table 1.

DNA extraction was performed by routine methods (phenol/chloroform); after precipitation, and final washing with ethanol, the dried pellets were re-suspended TE buffer, 50 μL aliquots and kept froze up to the time of using.

Whole-exome sequencing libraries, (genomic DNA, 1.5 µg) from peripheral whole blood, was prepared and analysed using a commercial target enrichment kit (Agilent SureSelectXT HumanAllExon V5+UTRs; Agilent Technologies, Santa Clara, CA, USA) and sequenced on a HiSeq1000 platform (paired-end 2 × 100 nt; Illumina, San Diego, CA, USA).

After quality-filtration, the reads were aligned to the reference human genome sequence (GRCh37/hg19) with an ISAAC aligner (Raczy, C. et al. Isaac: ultra-fast whole-genome secondary analysis on Illumina sequencing platforms. Bioinformatics 29, 2041–3, 2013). Variant annotation (limited to variants with a minimum quality score of 20 and a minimum read depth of 30×) was performed with VarSeq v1.4.5 (Golden Helix, Inc., Bozeman, MT, USA). Variants with a population frequency of > 5% (gnomAD) were excluded. For the variants analyzed, the effect on protein function (damaging or tolerated) was reported as in SIFT and Polyphen.

## 5. Discussion

### 5.1. Patient’s Phenotype

The dysmorphic features of the patient at birth, and shortly thereafter, were typical for CdC, suggesting the diagnosis. In the case report, we extensively included all available clinical data, because after the old description of the syndrome, it is uncommon to read detailed clinical reports of CdC cases in older ages. Such reports are needed to learn about clinical variability and to assess how common are clinical signs additional to “dysmorphic features, developmental delay, motor disturbances”. Some features (pain perception, PET analysis) were studied only in a few cases of CdC patients by our group, as per the detailed references quoted.

### 5.2. Esophageal Cancer

Esophageal cancer is classified by histology as adenocarcinoma (EAC) or squamous cell carcinoma (ESCC); EAC incidence has increased several folds in Western countries in recent years. It occurs predominantly in the lower esophagus, near the gastric junction, and is associated with a number of risk factors, including male sex, smoking, alcohol, obesity, advanced age, a clinical history of GER disease, and a history of a precursor state termed Barrett’s esophagus [13].

Based on Globocan 2020 [14], esophageal cancer is in the eighth place worldwide among the commonest cancer types. In Italy, similar data are available: in 2020, about 2400 new esophageal cancer cases have been diagnosed. The risk to develop esophageal cancer is clearly age-related; in fact, its rate × 100.000 is below 5 at age 40, steadily increases after age 44, and becomes more than 25 after 80 years [15].

The available data on cancer in CdC patients are limited to the paper by Guala et al. [6]. A likely explanation for the failure to identify other cases in the quoted collection of CdC cases [2,3,4,5] is that these reports include mainly pediatric or young patients, which is at variance with the paper based on Italian and German databases [6], in which the percentage of patients older than 40 years was 18.7%.

In discussing the relationship, if any, between CdC and esophageal cancer, we considered the presence of known risk factors, the results of all clinical investigations, and the results of molecular data.

The personal clinical history of our patient clearly discloses the presence of common clinical risk factors for EAC, such as male sex, obesity, GER since a young age, and Barret’s esophagus [13].

Obesity is not included in the description of the syndrome in Orphanet (updated 2020). It is noteworthy that the presence of GER in CdC patients is recorded both in the Italian and Danish register, with a similar prevalence, respectively, of 15 out of 119 (12.6%) and 5 out of 62 (8%) [2,16]. In addition, the family originates from an area (Lombardia) where the incidence of this cancer is slightly above the Italian national mean [14].

### 5.3. Pain Perception

Reduced or altered perception of pain is commonly reported for patients affected with developmental delay, but accurate neurophysiological studies are lacking. Painful syndromes and/or altered pain perception are frequently the result of a peripheral neuropathy with altered nerve conduction. In our case, nerve conduction was normal, but the quantitative sensory test (QST) showed an increased threshold of pain [10]. These data support a substantial integrity of type II Aβ fibers and a possible alteration of type III Aδ and type IV C fibers of the spinothalamic tract. Of course, in patients with CdCs, the altered response to pain might be the result of defects of central sensorial processing.

The altered pain perception demonstrated in our case is not associated with any anamnestic data related to the consumption of hot food and beverages, which is, however, a risk factor for squamous cell carcinoma but not (or in a very limited way) for esophageal adenocarcinoma [14].

It is more likely that the impaired ability to verbally express feelings of pain (esophageal and gastric pain in this case), along with the increased pain threshold, will result in reduced communication of “warning signals” and, in turn, a reduced or absent recognition of the symptom and its pharmacological management.

### 5.4. PET Analysis

The insula has been described to play a key role in brain processing of noxious and innocuous thermal stimuli. The anterior and the posterior portions of the insular cortex are involved in different ways in nociceptive and thermoceptive processing, and they are functionally connected to a large brain network involved in these functions, such as primary and secondary somatosensory cortices, anterior cingulate gyrus, prefrontal cortex, and parietal association cortices [17].

In our patient, no areas of significant reduction of metabolism were detected in either the insular or somatosensory cortex supporting the integrity of this central system function. Areas of hypometabolism were detected in the posterior cingulate cortex, inferior parietal gyrus, and dorsolateral frontal cortex. These areas represent novel findings, because the hypometabolic regions reported in the previous literature are the left temporal lobe, the right frontal subcallosal gyrus, the caudate body, the thalamus, the cerebellum, and the midbrain [12,18]. These areas’ findings may be correlated with some clinical features of this patient, such as the movement disorder with the cerebellar hypometabolic finding. The hypometabolism of the posterior cingulate cortex, inferior parietal gyrus, and dorsolateral frontal cortex have been shown to be involved in the frontoparietal network, which responds to planning, activity change, working memory, and executive attention [19], which were found to be impaired in this patient.

Interestingly, frontobasal areas and the caudate nucleus were also found to be hypometabolic in this patient. These areas appear to be involved in the control of impulsivity and in food cravings and eating disorders. Finally, it has been suggested that the latter condition may increase the risk of esophageal adenocarcinoma [20].

### 5.5. Molecular Investigations

For all genes discussed, the letters (p) or (a) refer to the presence or absence in the patient of a non-synonymous exonic variant (original data available upon request).

A general analysis of the results of the exome analysis of the trio performed to look for the variants present under the conventional models of inheritance (Table 1) failed to identify any gene that could be a strong and likely risk factor for esophageal cancer.

None of the variants listed in Table 1 have a strong or defined effect as a tumor cancer gene; some of the genes belong to pathways somehow related to many different types of cancer, but we did not identify any suggestion for variants acting as a risk factor for the early development of cancer.

In the medical literature, molecular analyses trying to correlate, in general, genes located in the short arm of chromosome 5 and cancer are numerous, but did not identify susceptibility variants with high clinical significance.

Looking for esophageal carcinoma susceptibility in OMIM, we found listed at #133239 *RNF6* and *DCC;* the latter, according to the STRING database [21], is linked by experiments/co-expression/co-occurrence to *TRIO* (*5p15.2*)(a) and *MYO10* (*5p15.1*)(p). Both genes are included in the region deleted in the patient. *MYO10* promotes tumor progression by inducing genomic instability, which, in turn, creates an immunogenic environment for immune checkpoint blockades; when upregulated in both human and mouse tumors, its expression level predisposes tumor progression [22]. The two *MYO10* exonic missense variants found in the patient, which are present as a single copy, are common, reported as benign, and thus unlikely to be relevant for the development of the disease.

In recent years, a large number of papers have discussed the relationship between genetic variants of esophageal cancer and esophageal carcinoma in particular. Among the numerous available papers, we focused our attention on the following: (i) Cancer Genome Atlas Research Network: Integrated genomic characterization of oesophageal carcinoma [13]; (ii) Dong et al.: Polygenic and epidemiological risk factors in EAC [23]; and (iii) Tian et al.: Association between genetic polymorphisms and esophageal cancer susceptibility [24].

The Cancer Genome Atlas Network reported significant mutations in genes *TP53*(p), *CDKN2A*(a), *ARID1A*(p), *SMAD4*(a), and *ERBB2*(a), which are not located on chromosome 5, and the only exonic missense mutation carried by the patient (*TP53*, pPro72arg) is interpreted as benign [13]. Dong et al. constructed a polygenic risk score (PRS) based on twenty-three previously published variants [23]; among them, only one is located in a gene located on chromosome 5, *TPPP* (5p15.33), which is not deleted in our case. None of the other variants listed in the PRS are present in the patient.

In the paper by Tian et al., who published a metanalysis of GWAS and esophageal cancer, variants in five genes were associated with EAC [24]; none of them are localized on chromosome 5p *(ERCC2*(a), *GSTP1*(p), *hOGG1*(p), *MTHFR*(p), and *CCND1*(a)), and the missense exonic variants of these genes present in our patient were always reported as benign.

## 6. Conclusions

We report a case of a patient with CdC who developed esophageal cancer at a very young age (29 years). We fully understand that some limitations are inherent to reporting single cases: the information obtained cannot be discussed generally, and there is a risk of overestimating the peculiar findings of that case. Nevertheless, in rare diseases for which each single center has a very limited chance to collect many similar, uncommon observations, the only way to add to the natural history of a severe illness is by reporting such cases. Grouping together several similar single reports will eventually result in an updated description of the disease.

No evidence was obtained through exome analysis for the presence of any variant, either under the model of monogenic disorders (Table 1) or as a variant already known, that is a risk factor for EAC.

This result is in keeping with the statement by Dong and coworkers, who found that the PRS they developed “increases discrimination and net reclassification of individuals with vs. without BE and EAC”, but, because of the small magnitude of the improvement, is not of clinical use [23].

Our study, after extensive clinical and molecular work, suggests that the presence of well-known risk factors (maleness, obesity, and GER disease) in a patient with very limited capability of expressing discomfort or referring clinical symptoms have been the main risk factors for developing the EAC.

In addition, in this patient, the pain threshold was very high [11], and, in fact, even during cancer therapies, he never complained of any pain; the mother only reported weight loss, tiredness, and fatigue. On the other hand, no alterations were found on PET in the primary sensory cortex or in the insula.

In conclusion, based on the published data and the unpublished information provided by some family association [2,3,4,5], there is no evidence for any increased risk of developing cancer in CdC patients. The few reported cases (papillary thyroid cancer (ptc), gastric carcinoid tumor (gct), and breast cancer (bc)) [6] may thus be anecdotical.

However, the fact that most cases showed an early age of onset (cancer developed, respectively: ptc at 20 yrs (general population, 30–50), gct at 10 yrs (general population, 55–65), bc at 50 yrs (mean for general population, 62), and a second case of breast cancer with a strong positive family history developed at 31 yrs), confirmed also in this case of esophageal cancer, is intriguing. In fact, early onset of cancer is a common hallmark of genetically determined cancer. Reporting similar cases is thus worthwhile. In a similar chromosomal syndrome, Wolf–Hirschhorn (4p-) (WHS), an increasing number of hepatic neoplasms has been reported, and the authors suggest that the tumor “may be a feature of WHS” [25].

The comparison between molecular data in different patients might disclose molecular abnormalities in similar metabolic pathways related to cancer development. Likely, many variants with minor effects may contribute, and their identification requires more cases and the use of new tools for bioinformatic analysis.

Future perspectives for early diagnosis of this type of cancer might include extended whole genome analysis, including deep intronic or regulatory variants in other genes.

In addition, epigenetic studies [26,27] are likely to be relevant, as they are now opening new insights into many different environmental factors acting on physiological development, which may possibly be relevant for subjects showing developmental problems.

For the time being, we emphasize that any clinical signs reported or observed in a patient with very limited ability to express themselves, such as CdC patients, must never be underestimated and require full clinical investigation.

## Figures and Tables

**Figure 1 diseases-12-00009-f001:**
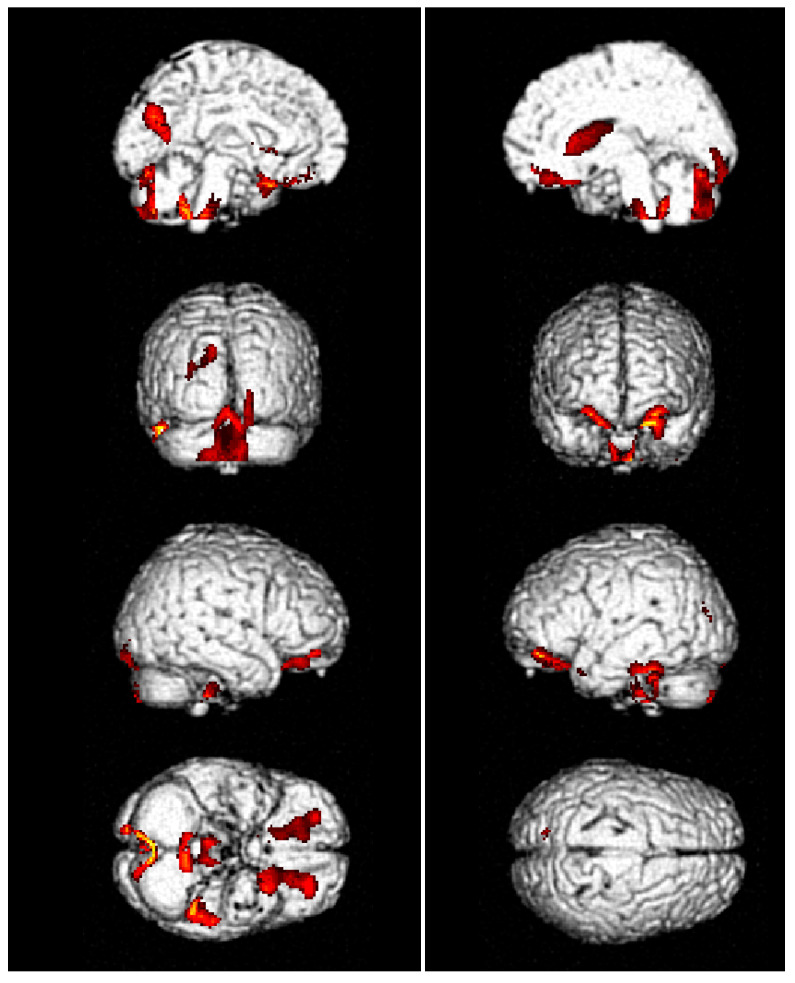
PET imaging. Brain three-dimensional rendering showing the hypometabolic regions in the CdC patient compared to controls. Areas of significant hypometabolism (in red) were found in the left precuneal region and posterior cingulate cortex, in the fusiform gyrus, in the right body of the caudate nucleus, and in the posterior cerebellar lobe bilaterally. Other areas were found in the frontal–orbital cortex, bilaterally. No hypometabolic area was detected in the somatosensory or insular cortex.

**Figure 2 diseases-12-00009-f002:**
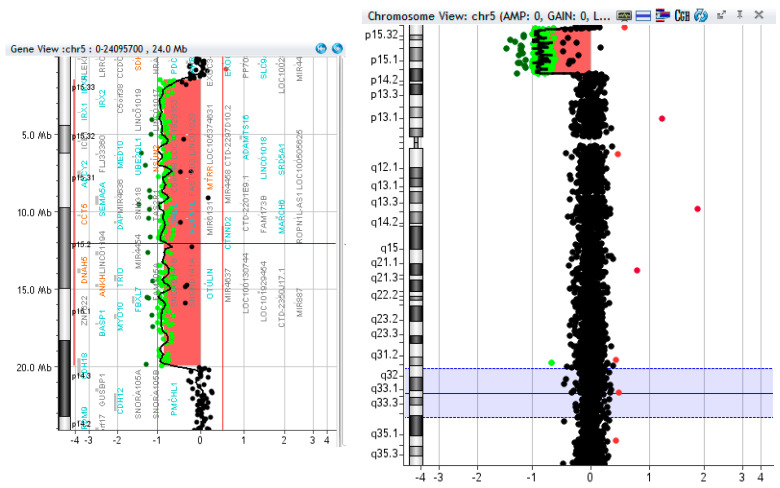
Oligo array-CGH log2 ratio plot of whole chromosome 5 with ideogram showing deletion.

**Table 1 diseases-12-00009-t001:** Rare variants found in the patient and filtered under models of autosomal recessive inheritance (homozygous or compound heterozygous, Cpd Het) and de novo variants.

Gene	Genotype	Variant ID	Protein Change	Allele Frequency	Mutation Type	ENST
AKAP13	Cpd Het	15-85581589-C-G	p.Ala1174Gly	0.00003287	Missense	ENST00000361243
	Cpd Het	15-85710582-C-T	p.Pro1850Ser	0.0005716	Missense	ENST00000361243
	Cpd Het	15-85723260-C-T	p.Arg2233Trp	0.0001051	Missense	ENST00000361243
APOB	Cpd Het	2-21002134-A-T	p.Ser4430Thr	0.00007885	Missense	ENST00000233242
	Cpd Het	2-21005467-A-T	p.Ser3801Thr	0.001243	Missense	ENST00000233242
SPATA31E1	Cpd Het	9-87887717-CA-C	p.Ala1078ArgfsTer15	0.000006569	Frameshift	ENST00000325643
	Cpd Het	9-87888455-T-C	p.Val1323Ala	0.0002563	Missense	ENST00000325643
ATG9B	de novo	7-151019011-CCAG-C	-	-	Deletion	ENST00000639579
BICRA	de novo	19-47679481-TC-T	-	-	Frameshift	ENST00000396720
DGAT1	de novo	8-144317787-C-A	p.Gln297His	-	Missense	ENST00000528718
DNAH1	de novo	3-52347826-C-A	p.Pro653His	-	Missense	ENST00000420323
GHDC	de novo	17-42193541-GGCA-G	p.Leu13del	0.0001974	Deletion	ENST00000301671
MDM1	de novo	12-68296985-TAA-TA	-	-	Splice region variant, Intron variant	ENST00000303145
NAV1	de novo	1-201782201-C-A	p.Asp172Glu	-	Missense	ENST00000367296
PROSER3	de novo	19-35762152-TC-T		-	Splice region variant, Intron variant	ENST00000396908
GRIPAP1	de novo	X-48981848-CCTG-C	p.Gln541del	0.000	Deletion	ENST00000376423
PLXNB3	de novo	X-153771948-C-A	.	-	Missense	ENST00000538966
SPTBN5	de novo	15-41862648-C-A	p.Glu2426Ter	-	Nonsense	ENST00000320955
UBR4	de novo	1-19210192-CG-C	.	-	Frameshift	ENST00000375254
ZNF579	de novo	19-55578712-C-A	p.Ala310Ser		Missense	ENST00000325421
KLK5	Recessive	19-50950027-C-T	p.Gly55Arg	0.006245	Missense	ENST00000336334

## Data Availability

Full details about clinical data and all molecular data are available on request; Authors producing these data agrre on this statement.

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
