# Peer review of "Esophageal Cancer with Early Onset in a Patient with Cri du Chat Syndrome"

_diseases, 2023, doi:10.3390/diseases12010009_

Round 1
Reviewer 1 Report
Comments and Suggestions for Authors
Dear Authors,
I am glad to read your MS entitled "Esophageal Cancer with Early Onset in a Patient with Cri Du Chat Syndrome" by Danesino and colleagues. It is pretty inetersting to show " the psossibility" of a potential problem in a rare diseas such as Cri du Chat, even before it will be a public health problem for those patients. A big challenge for the researchers is the scarce data regarding adults and longitudinal problems in rare disease were most of the cases published are pediatrics. So thanks´ you in advance for taking this face.
However some minor comments have to be done:
1.- Please I think that the clinical history may be improved if you tell us as a longitudinal study in a table. It is a suggestion.
2.- I miss more details in description joined to your new patient from the other previous one, at least in the discussion just to compare here..... at least age of onset of cancer, type etc.........
3.- page 3 line 143, why 18f-flurodesoxyglucose are in bold and upper size of the letter? please explain if it has any explanation or importance.
4.- Again in the disucssion i miss to crate a temporal description with more relevant longitudinal episodes in the patient to be discuss with other previous patients, just to try to see a putative relatioship between cancer and adult CdC patients, as it will be ocurre in WHS and Hepatomes during the adolescence or young teen agers, some comments should be introduce them in the discussion
5.- why page 8 lines 308-310 are in bold??
6.- I think a potential comment regarding future whole genome sequencing in the patients should be included, regarding the possibility of other variants, such as deep intronic or regulatory to manage the cancer in the patient.
Author Response
Reviewer 1
We appreciated the first sentence in the comments of Rev.1 as it demonstrates as he fully recognize the aims of our work: to present a case report, which might in the future be relevant for a better understanting clinical problems in Cri du Chat syndrome.
All changes requested/suggested are in red in the revised manuscript
Comment 1- At present, we did not summarize clinical data in a table, because we feel that a full text “case report” better describes the patient, his clinical history and all the tests performed.
Comment 2- according to suggestion, we added the information as to the type of tumor and age of onset in CdC patient.
Comment 3- misprint, we changed
Comment 4-we added a comment about WHS, as requested
Comment 5- misprint, we changed
Comment 6- modified as suggested
Reviewer 2 Report
Comments and Suggestions for Authors
The study's exploration of the infrequent occurrence of cancer as a comorbidity in Cri du Chat (CdC) patients is intriguing. However, the constrained sample size, particularly the limited number of reported cancer cases in Italy and Germany, introduces concerns regarding the applicability of the study's conclusions. Furthermore, the absence of documented cancer cases in several other databases undermines the significance of the findings. Additionally, the study's conclusion seems to derive mainly from a singular case, and the failure to identify genetic variants as potential risk factors for esophageal adenocarcinoma (EAC) in the exome analysis raises doubts about the study's overall robustness. Upon evaluation, I recommend rejecting the manuscript in its current state as it does not meet the criteria for publication.
Major Comments
11. The study reports a rare occurrence of cancer as a comorbidity in Cri du Chat (CdC) patients, but the small number of cases with cancer reported in Italy and Germany (4 out of 321 CdCs) raises concerns about the generalizability of the conclusions.
22. The absence of reported cancer cases in databases from Denmark, Spain, Australia and New Zealand, and Japan contradicts the significance of the findings in Italy and Germany, and questions the consistency of the observed phenomenon.
33. The study appears to rely heavily on a single case – a 29-year-old CdC patient with esophageal adenocarcinoma (EAC). Such a limited focus may not provide a comprehensive understanding of the relationship between CdC and cancer.
44. The failure to identify variants in the exome analysis as likely risk factors for EAC in the trio (patient and parents) raises concerns about the genetic basis of the observed cancer and the robustness of the study's conclusions.
5. 5. The conclusion drawn regarding the main risk factors for developing EAC (maleness, obesity, gastroesophageal reflux, and Barrett’s metaplasia) is based on a single case and may not be applicable to the broader CdC population.
Comments on the Quality of English Language
Minor editing of English language required
Author Response
Reviewer 2
In red our comments
Major Comments
- The study reports a rare occurrence of cancer as a comorbidity in Cri du Chat (CdC) patients, but the small number of cases with cancer reported in Italy and Germany (4 out of 321 CdCs) raises concerns about the generalizability of the conclusions.
We do not claim that cancer is a general feature or comorbidity of CdC; we only feel that observing, in the submitted “case report” a case of a tumor exceedingly rare at that age in a rare syndrome deserves to be reported. Quoting the few previous data available is a must in writing a paper.
- The absence of reported cancer cases in databases from Denmark, Spain, Australia and New Zealand, and Japan contradicts the significance of the findings in Italy and Germany, and questions the consistency of the observed phenomenon.
As, as stated before, we wish to add a “case report” to the existing ones, we tried to check as much as possible if and which similar observations have been made; In the paper about Italy and Germany, it has stated that the different ages in the available samples, older in Italy and Germany, much younger in other papers, may account for the absence of cases in other reports. For instance, in the review about CdC in Orphanet, 2006, no cancer is reported, likely because data were obtained from cases of pediatric age. If we do not report “case reports” with uncommon findings, we will never add to the evolving clinical picture of rare syndromes.
- The study appears to rely heavily on a single case – a 29-year-old CdC patient with esophageal adenocarcinoma (EAC). Such a limited focus may not provide a comprehensive understanding of the relationship between CdC and cancer.
Yes, it is a case report. We did not mean to solve the problem of the relationship between CdC and cancer, if any, but we intended only to report a new case of cancer in a rare syndrome. An additional and relevant reason to report such cases, even single cases, is that the age of the population of CdC patients is growing (updated information shows that 44 out of 175 Italian cases entered in the Database of the Family Association A.B.C. , are above 30 years of age) and thus at risk for age related pathologies including cancer.
- The failure to identify variants in the exome analysis as likely risk factors for EAC in the trio (patient and parents) raises concerns about the genetic basis of the observed cancer and the robustness of the study's conclusions.
The failure to identify genetic variants with a clear pathogenic significance, is relevant to suggest, as we did, that the impairment to express discomfort and reduced pain sensitivity over imposed to the well-known clinical risk factors (sex,gastric reflux, Barrett esophagus) are likely to have the been the main causes for developing esophageal carcinoma. Thus, in our opinion it does not decrease the robustness of the study, but helps to formulate our hypothesis.
55 The conclusion drawn regarding the main risk factors for developing EAC (maleness, obesity, gastroesophageal reflux, and Barrett’s metaplasia) is based on a single case and may not be applicable to the broader CdC population.
Yes, it is a case report, which shows all the main risk factors for developing esophageal carcinoma; as it appears in a case with a genetic syndrome, and it develops at a very young age ( a clinical feature observed in genetically determined cancers) it was , according to us, just mandatory to evaluate possible genetic causes.
WE ADDED IN THE PAPER (in red) A SENTENCE ABOUT THE LIMITATION OF THE STUDY.
Round 2
Reviewer 2 Report
Comments and Suggestions for Authors
The research seems to heavily depend on a solitary case, specifically that of a 29-year-old patient diagnosed with esophageal adenocarcinoma (EAC) related to CdC. This narrow focus may hinder the attainment of a comprehensive understanding of the association between CdC and cancer. The editor-in-chief may make a decision based on the journal's objectives and scope.
Comments on the Quality of English LanguageMinor editing of English language required